# Cross-Sectional, Short-, Medium-, and Long-Term Effects of Dietary Pattern on Frailty in Taiwan

**DOI:** 10.3390/ijerph18189717

**Published:** 2021-09-15

**Authors:** Mei-Huey Shiau, Meng-Chih Lee, Fang-Ling Lin, Baai-Shyun Hurng, Chih-Jung Yeh

**Affiliations:** 1Health Promotion Administration, Ministry of Health and Welfare, Taipei 103205, Taiwan; smeihuey2000@gmail.com; 2Department of Public Health, Chung Shan Medical University, Taichung 40201, Taiwan; flin11111@gmail.com (F.-L.L.); hbskimo@yahoo.com.tw (B.-S.H.); 3Department of Family Medicine, Taichung Hospital, Ministry of Health and Welfare, Taichung 40343, Taiwan; mengchihlee@gmail.com; 4Institute of Population Health Sciences, National Health Research Institutes, Miaoli 35053, Taiwan

**Keywords:** dietary pattern, elderly, frailty, reduced-rank regression, Taiwan

## Abstract

This study examined the association between dietary patterns and the development of frailty during 4-, 8-, 12-year follow-up periods in the population-based Taiwan Study. We used the data of an elderly population aged 53 years and over (*n* = 3486) from four waves of the Taiwan Longitudinal Study on Aging. Frailty was identified by using the modified Fried criteria and the values were summed to derive a frailty score. We applied reduced rank regression to determine dietary patterns, which were divided into tertiles (healthy, general, and unhealthy dietary pattern). We used multinomial logistic regression models to assess the association between dietary patterns and the risk of frailty. The healthy dietary pattern was characterized by a higher intake of antioxidant drinks (tea), energy-rich foods (carbohydrates, e.g., rice, noodles), protein-rich foods (fish, meat, seafood, and eggs), and phytonutrient-rich foods (fruit and dark green vegetables). Compared with the healthy pattern, the unhealthy dietary pattern showed significant cross-sectional, short-term, medium-term, and long-term associations with a higher prevalence of frailty (odds ratios (OR) 2.74; 95% confidence interval (CI) 1.94–3.87, OR 2.55; 95% CI 1.67–3.88, OR 1.66; 95% CI 1.07–2.57, and OR 2.35; 95% CI 1.27–4.34, respectively). Our findings support recommendations to increase the intake of antioxidant drinks, energy-rich foods, protein-rich foods, and phytonutrient-rich foods, which were associated with a non-frail status. This healthy dietary pattern can help prevent frailty over time in elderly people.

## 1. Introduction

Frailty is age-related and primarily characterized by decreases in functional reserves across multiple physiological systems. Frailty is highly prevalent in old age and engenders a high risk of falls, disability, hospitalization, and mortality [1]. Fried et al. defined the most frequently used criteria for identifying physical frailty; they implemented a standardized five-item index for determining frailty status, and the five items are as follows: unintentional weight loss, exhaustion, low physical activity, slowness, and low grip strength [1].

Nutritional status plays a major role in improving health and could be a crucial factor for successful aging, and poor nutrient intake may accelerate the transition from vulnerability to frailty and dependence [2,3,4]. Several studies have indicated that macronutrient and micronutrient intake or supplements are associated with reduced frailty levels. For example, a higher intake of certain micronutrients (vitamins A, C, D, B6, carotenoids, and folate) was associated with a lower prevalence of frailty [5,6], a higher intake of proteins or amino acids was associated with a lower risk of frailty [7,8], and a higher intake of fiber and carbohydrates reduced frailty status [9,10]. High protein capacity, high dietary total antioxidant capacity (such as from vegetables and green tea), and a combination of these were inversely associated with the prevalence of frailty among older Japanese women [11]. Systematic reviews and meta-analyses have suggested that a healthier diet has some beneficial effects against frailty [4,6,7,8,10]. It would seem that the main cause of frailty may be the inadequate intake of multiple nutrients and low food consumption. However, people do not eat meals with single nutrients in daily life; instead, they eat meals with a wide variety of nutrients. Food and nutrient intake is associated with certain dietary patterns. In European countries, a few studies have demonstrated that the Mediterranean dietary pattern reduced the risk of frailty among older people [12,13]. In addition, the consumption of pasta and biscuits and snacking patterns increased the risk of frailty [14].

Nevertheless, traditional Asian diets differ considerably from Western diets. Yokoyama et al. reported that frequently following the Japanese dietary pattern composed of a staple food, main dish, and side dish twice daily reduced the risk of pre-frailty and frailty [15]. Lo et al. conducted a cross-sectional study to investigate the association between dietary patterns and frailty and discovered that consuming phytonutrient-rich plant foods, tea, protein-rich foods, and omega-3-rich deep-sea fish was associated with a low frailty index in Taiwan [16]. However, longitudinal studies investigating the association between dietary patterns and changes in frailty over time have yet to be performed in Taiwan. To fill this gap in the literature, we conducted the present study to identify frailty-related dietary patterns and their short-term, medium-term, and long-term effects on frailty development over a 12-year follow-up period in older adults (aged ≥ 53 years) selected from the Taiwan Longitudinal Study on Aging (TLSA).

## 2. Materials and Methods

### 2.1. Study Design and Sampling

We used data from four waves of the TLSA (1999, 2003, 2007, and 2011), which is a nationally representative study of adults aged ≥ 60 years in 1989, with younger refresher cohorts added in 1996 and 2003 to maintain and extend the representativeness of the sample to the population aged ≥ 50 years. Trained interviewers administered face-to-face interview questionnaires. A more detailed description of the TLSA has been provided elsewhere [17,18]. In brief, the survey provided information pertaining to the socioeconomic status; demographics, lifestyles, and health factors of older adults in Taiwan. The TLSA was approved by the Research Ethics Committee of the Bureau of Health Promotion, Department of Health, Taiwan (10000800524). All participants signed an informed consent form before the interviews.

We identified 3945 participants with available data on dietary intake and frailty status. We excluded participants with missing data on demographic characteristics and anthropometrics (*n* = 459), leaving an effective sample of 3486 participants aged ≥53 years in 1999 for analysis.

### 2.2. Questionnaire

Standardized face-to-face interviews were conducted to gather information on the participants’ sociodemographic characteristics (age, sex, education level, ethnicity, and financial status), lifestyles (current smoking habits, alcohol intake, and exercise), health status (number of diseases), mobility, and dietary characteristics (e.g., intake frequency).

We identified and counted the number of participants who self-reported having the following 10 chronic diseases: hypertension, diabetes, heart disease, stroke, cancer, respiratory disease, arthritis, liver disease, gastrointestinal disorders, and kidney disease.

We assessed an individual’s mobility level as their capacity to execute the following six activities: stand continuously for 15 min, squat, raise both hands over their head, grasp or turn objects with their fingers, run a short distance (20 to 30 m), and walk up two or three flights of stairs. Each activity was scored from 0 (“no difficulty”) to 1 (“mild to severe difficulty”). Participants with a total score of 0 were assigned to the “no-problem” group, and those with a total score great than 0 were assigned to the “impairment” group.

### 2.3. Dietary Assessment

The dietary assessment questionnaire was adapted from the Mini Nutritional Assessment, which was developed by Guigoz et al. [19]. After being translated and adapted into Taiwanese, the dietary questionnaire was modified on the basis of Taiwanese cultural considerations [20]. For participants from the TLSA cohort who were interviewed on their diet in 1999, dietary intake was assessed using modified food frequency questionnaires including 11 primary food items: tea, fruit, fish, meat/poultry, seafood (fish not included), dark green vegetables, eggs, other vegetables, milk, rice or noodles, and beans. Dietary characteristics included appetite, changes in intake amount, reduced intake due to indigestion, and reduced intake due to disease.

### 2.4. Anthropometric Measurements

We included the following anthropometric measurements in this study: self-reported body weight (kg) and height (cm). A trained interviewer measured mid-arm circumference and calf circumference according to standard operating procedures by using a flexible but non-stretchable measuring tape. Body mass index (BMI) was calculated as body weight (kg) divided by height squared (m^2^).

### 2.5. Frailty

Frailty was measured on the basis of the following modified Fried frailty criteria: shrinking (self-reported poor appetite occurring often or most of the time during the past week), exhaustion (agreement with the statement “I could not get going” or “I felt everything I did was an effort” often or most of the time during the past week), weakness (having difficulty or being unable to carry 12 kg objects), slow walking speed/slowness (having difficulty or being unable to walk a distance of 200 to 300 m), and low physical activity (high physical activity, comprising those who gardened, took walks, jogged, climbed mountains, or engaged in other outdoor activities at least once or twice a week; and low physical activity, comprising those who did not engage in the aforementioned activities). Frailty scores ranged from 0 to 5. Participants were classified as frail if they met three or more criteria, pre-frail if they met one or two criteria, and non-frail if they met no criteria [21].

### 2.6. Potential Confounders

At baseline, information was gathered on sociodemographic (age, sex, educational level, ethnicity, family income), lifestyle (smoking and drinking habits), anthropometric (BMI, mid-arm circumference, calf circumference, and leg length), dietary (daily meal frequency; food amount; appetite; whether the participant ate alone; changes in food in-take amount; whether the participant ate less from indigestion, constipation or diarrhea; whether the participant ate less because of diseases or medical orders), and disease (number of major diseases) variables. These variables could act as confounders in the current study because of their relationship with both dietary pattern and frailty.

### 2.7. Statistical Methods

We analyzed data using the SAS Statistical package (version 9.4, SAS Institute Inc., Cary, NC, USA). We applied reduced-rank regression (RRR) [22] to derive dietary patterns from the TLSA 1999 data concerning the 11 main food items. In this study, we conducted RRR analysis on 1689 subjects with complete frailty score data at four time points. Statistical methods for dietary pattern analysis, such as factor analysis [23] and exploratory principal component analysis (PCA) [24,25], may or may not be associated with the outcome of interest. Moreover, such methods may not be able to derive dietary patterns that are predictors of disease. This study identified dietary patterns using RRR, which reduces the dimension of predictor variables (food frequencies) and maximizes the variation explained by the response variable (frailty score). Tea, fruit, fish, meat/poultry, seafood, eggs, vegetables, milk, and beans constituted some of the food items considered in this study, and we calculated the consumption frequency of each of these items by using a score from 0 to 7 (0 = “never”, 0.5 = “less than 1 time per week”, 1.5 = “1 to 2 times per week”, 4 = “3 to 5 times per week”, and 7 = “almost every day”). We also calculated the consumption frequency of dark green vegetables by using a score from 1 to 7 (1 = “less than 2 times per week”, 4 = “3 to 5 times per week”, and 7 = “almost every day”), and the daily consumption of rice or noodles by using a score from 1 to 5 (1 = “1 bowl or less than 1 bowl per day”, 2 =“ 2 to 3 bowls per day”, 3 = “4 to 5 bowls per day”, 4 = “6 to 7 bowls per day”, and 5 = “8 to 9 bowls per day”). All food item scores were used to estimate dietary pattern scores.

In the univariate analysis of the second dietary pattern, we observed that three tertile groups were not associated with frailty status in 1999, 2003, 2007, or 2011; therefore, we used only the first dietary pattern for further analysis.

We conducted a chi-square test and analysis of variance to test the association of frailty status with sociodemographic characteristics, behavioral variables, mobility, anthropometric measurements, number of diseases, and dietary characteristics at the baseline year (1999). We applied multiple multinomial logistic regression to explore the associations between dietary patterns and frailty status in 1999, 2003, 2007, and 2011. After excluding those with frailty in the preceding years, we fitted 2474, 1913, and 879 participants into the models for our short-term (2003), medium-term (2007), and long-term (2011) association analyses, respectively. Four multiple multinomial logistic regression models were built. Model 1 was adjusted for demographic and lifestyle variables, model 2 was adjusted for adjusted for variables in Model 1 and added anthropometric variables from 1999, model 3 was adjusted for variables in variables in Model 1 and added dietary characteristic variables from 1999, and Model 4 was adjusted for variables adjusted for variables in Models 2 and 3 and added the variable concerning number of diseases. Associations between dietary patterns and frailty status were investigated separately for men and women. Because most results were similar between the sexes, we only report the associations between dietary patterns and frailty status for the entire study population.

## 3. Results

### 3.1. Dietary Patterns and Factor Loading Values

Table 1 presents the factor loadings derived for the 11 food items in two dietary patterns. All the factor loadings of the food items in the first dietary pattern were negative. More negative factor loading values for the food items were associated with lower frailty scores. The percent variation accounted for by RRR on factor 1 (first dietary pattern)and food items (model effects) was 15.10%, and frailty scores (response variables) was 8.02%. On the basis of RRR scores of the first dietary pattern, we divided continuous dietary patterns into three tertiles: The first tertile comprised those with low scores, the second tertile comprised those with intermediate scores, and the third tertile comprised those with high scores. This dietary pattern was characterized by the consumption of tea (−0.46), carbohydrates (−0.41), fruit (−0.40), fish (−0.35), meat/poultry (−0.33), seafood (−0.27), eggs (−0.23), dark green vegetables (−0.21), other vegetables (−0.18), milk (−0.12), and beans (0.11), according to the absolute values of the factor loadings. On the basis of the continuous dietary pattern scores, participants in the first tertile, the second tertile, and the third tertile had a healthy dietary pattern, general dietary pattern, and unhealthy dietary pattern, respectively. The healthy dietary pattern was characterized by higher intake of antioxidant drinks (tea), energy-rich foods (carbohydrates, e.g., rice, noodles), protein-rich foods (fish, meat, seafood, and eggs), and phytonutrient-rich foods (fruit and dark green vegetables). The general dietary pattern and unhealthy dietary pattern were characterized by medium and lower intake of certain food items which have been mentioned above.

### 3.2. Participants’ Characteristics

Table 2 presents the baseline (1999) characteristics of participants stratified by frailty status. Overall, 474 (13.6%), 1512 (43.4%), and 1500 (43.0%) participants were classified as frail, pre-frail, and non-frail in 1999. Moreover, 9.3% of men and 18.8% of women had frailty. Participants with frailty tended to be older, have lower educational attainment, be part of the Fujian ethnic group, and have unsatisfactory income compared with participants without frailty. Those with frailty exercised less frequently, did not have smoking or drinking habits at the time of the study, had a lower BMI, had a smaller upper arm circumference, had a smaller leg circumference, had a shorter leg length, had more impaired mobility function, and had more diseases.

### 3.3. Dietary Characteristics Stratified According to Frailty Status

Table 3 lists the baseline (1999) dietary characteristics of participants stratified by frailty status. Participants with frailty had the lowest percentage of daily meals ≥ 3 meals (92.6%), intake enough food (97.9%), good appetite (55.1%), intake amount no change (77.0); and the highest percentage of eat alone (21.7%), had reduced food intake due to indigestion (25.1%), and reduced food intake due to disease (34.6%).

### 3.4. Dietary Patterns and Frailty Status

Table 4 presents the associations between the tertiles of dietary patterns and frailty status as well as the cross-sectional (1999), short-term (2003), medium-term (2007), and long-term (2011) associations with frailty status. We applied 4 models to adjust the confounding covariates: Model 1 adjusted for demographic and lifestyle variables, including age, gender, education, ethnicity, income/current economic status, current smoking habits, and current alcohol use. Model 2 adjusted for variables in Model 1 and added anthropometric variables from 1999, including BMI, mid-arm circumference, calf circumference, and leg length. Model 3 adjusted for variables in Model 1 and added dietary characteristic variables from 1999, including daily meal frequency; adequacy of food amount; appetite; whether the participant ate alone; changes in food intake amount; whether the participant ate less from indigestion, constipation, or diarrhea; and whether the participant ate less from diseases or medical orders. Model 4 adjusted for variables in Models 2 and 3 and added the variable concerning number of diseases. In an analysis of cross-sectional (1999) associations, compared with the group with a healthy diet, the group with an unhealthy diet had a significantly higher prevalence of frailty and pre-frailty (ORs = 4.28 (95% CI 3.10–5.90), 4.13 (95% CI 2.99–5.71), 2.78 (95% CI 1.97–3.93), and 2.74 (95% CI 1.94–3.87) for frailty and ORs = 1.59 (95% CI 1.31–1.94), 1.56 (95% CI 1.28–1.89), 1.42 (95% CI 1.16–1.74), 1.41 (95% CI 1.15–1.72) for pre-frailty in Model 1, 2, 3, and 4, respectively). Compared with the group with a healthy diet, the general dietary group had a significantly higher prevalence of frailty and pre-frailty (ORs = 1.78 (95% CI 1.27–2.50), 1.80 (95% CI 1.29–2.52), 1.55 (95% CI 1.09–2.21), and 1.55 (95% CI 1.09–2.21) for frailty and 1.32 (95% CI 1.10–1.58), 1.31 (95% CI 1.10–1.57), 1.26 (95% CI 1.05–1.51), and 1.26 (95% CI 1.05–1.51) for pre-frailty in Model 1, 2 3, and 4 respectively), although the ORs were lower than those for the comparison between the healthy and unhealthy group.

After excluding those with frailty in the preceding years, we fitted 2474, 1913, and 879 participants into the models for our short-term (2003), medium-term (2007), and long-term (2011) association analyses, respectively. For the short-term association analysis, Model 4 indicated that the unhealthy dietary group was significantly associated with a high incidence of frailty (OR = 2.55; 95% CI 1.67–3.88) but that it was not significantly associated with the incidence of pre-frailty (OR = 1.26; 95% CI 1.00–1.60). The general dietary group had a significantly higher incidence of frailty and pre-frailty (OR = 1.78; 95% CI 1.18–2.69) for frailty and OR = 1.23; 95% CI 1.00–1.52 for pre-frailty). In the medium-term association analysis, Model 4 revealed that the unhealthy dietary group was significantly associated with a high incidence of frailty (OR = 1.66; 95% CI 1.07–2.57) but that it was not significantly associated with the incidence of pre-frailty (OR = 1.36; 95% CI 1.03–1.80). We observed no significant association between the general dietary group and the incidence of frailty or pre-frailty (OR = 1.37; 95% CI 0.93–2.03 for frailty and OR = 1.05; 95% CI 0.83–1.33 for pre-frailty). For long-term association analysis, the unhealthy dietary group was significantly associated with a high incidence of frailty and pre-frailty in full Model 4 (OR = 2.35; 95% CI 1.27–4.34 for frailty and OR = 1.81; 95% CI 1.20–2.74 for pre-frailty); however, we observed no significant association between the general dietary group and the incidence of frailty or pre-frailty (OR = 1.17; 95% CI 0.67–2.05 for frailty and OR = 1.26; 95% CI 0.90–1.78 for pre-frailty).

### 3.5. Sociodemographic, Behavioral, Dietary, Anthropometric, Health Status, and Food Frequency Characteristics among Three Dietary Groups

Table 5 presents participants’ characteristics stratified by dietary pattern score tertile. Participants with a healthy dietary pattern tended to be male, younger, have a higher education level, have a satisfactory income level, have smoking or drinking habits at baseline, exercise more frequently, and have no mobility problems. They had a good appetite, had no change in food intake amount, did not report reduced intake due to indigestion, had a higher BMI and leg circumference, and had fewer numbers of diseases. Regarding food consumption frequency, participants with a healthy dietary pattern likely had a higher frequency of daily meals and consumed more meat, seafood, eggs, milk, beans, vegetables, fruit, tea, and dark green vegetables than did those with a general dietary pattern or unhealthy dietary pattern.

## 4. Discussion

By applying RRR, we identified a healthy dietary pattern exhibiting an inverse dose-response association with frailty in Taiwanese community-dwelling older people. We observed that lower consumption of the food in this healthy dietary pattern was longitudinally associated with a higher risk of frailty during follow-up periods of 4, 8, and 12 years. This healthy dietary pattern comprised antioxidants (tea), carbohydrates (rice), protein-rich foods (fish, meat, seafood, eggs, and milk), and phytonutrient-rich foods (fruit, dark green vegetables, other vegetables, and beans). Our study is the first to investigate the longitudinal association between dietary patterns and frailty in a community-dwelling older Taiwanese population by using an empirical dietary pattern method.

According to our review of the literature, only three population-based studies have applied dimension reduction analysis to derive dietary patterns and longitudinally examine the relationships between dietary patterns and frailty [23,24,25]. A Spanish prospective study [23] of 1872 individuals aged 60 years indicated that a prudent dietary pattern characterized by a high intake of olive oil and vegetables showed an inverse dose-response relationship with frailty incidence over a 3.5-year follow-up. By contrast, a Westernized pattern characterized by a high intake of refined bread, whole dairy products, and red and processed meat had a direct relationship with an increased risk of slow walking speed and weight loss. A prospective study [24] conducted on 2632 individuals aged 45 years in the Netherlands revealed that a traditional dietary pattern characterized by high consumption of legumes, eggs, and savory snacks was associated with a lower incidence frailty over a 4-year follow-up. By contrast, a carnivore pattern comprising high meat and poultry consumption was significantly associated with an increased frailty index over time; however, the association became non significant after adjustment for energy intake. In cross-sectional analyses, adherence to these patterns was not associated with frailty. In a Hong Kong prospective investigation [25] of 2724 Chinese elderly participants, no association was observed between the “vegetable–fruit” or “meat-fish” dietary pattern identified and incident frailty; a higher diet quality was associated with a lower risk of frailty during a 4-year follow-up period. In this Hong Kong study, the three dietary patterns did not appear to offer significant protective effects against frailty based on multivariate adjusted ORs. If these three dietary patterns were to be integrated into one pattern, this combined pattern would have similar characteristics to the RRR-derived dietary pattern in this study. The protective effects of dietary patterns in that Hong Kong study were dispersed by three factors, which may have resulted in non-significant findings. In the current study, we combined these three factors using RRR to highlight the association between dietary pattern and frailty. It would be worthwhile to conduct further studies to verify these posteriori methods.

Because of differences in the definitions of frailty, dietary patterns, and covariates between our study and the aforementioned studies, directly comparing our observations with published findings would be challenging. Overall, our study findings are consistent with those of the prospective studies in Spain and the Netherlands. However, our findings are not consistent with those of the Hong Kong study, and this inconsistency may be attributed to the differences in the statistical methods used to generate dietary patterns (RRR or principal component analysis) between the two studies.

Green and black tea are popular drinks in Taiwan. In this study, tea was the most important item in the frailty-related dietary patterns. Previous studies have indicated that oxidative stress and inflammation may play a major role in the development of frailty [26,27]. Polyphenolic fractions isolated from green tea inhibit oxidative stress and maintain anti-inflammatory processes, resulting in strong plasma antioxidant activity [28,29].

Frailty is inversely associated with high consumption of vitamin A, carotenoids, cryptoxanthin, vitamin D, α-tocopherol, vitamin B6, folate, vitamin C, selenium, and vitamin E [30,31]. These nutrients are abundant in vegetables, dark green vegetables, fruits, and tea.

Older adults have a high risk of inadequate protein intake. Inadequate protein intake may engender loss of muscle mass and strength, eventually leading to sarcopenia and frailty [32,33,34]. Adequate intake of essential amino acids and carbohydrates can prevent muscle protein loss during bed rest [35]. High intake of meat, dairy products, and animal and plant proteins is generally associated with a low incidence of frailty [36,37]. A previous study reported that low intake of energy daily and low intake of more than three nutrients were significantly and independently associated with frailty [27].

A previous Taiwanese cross-sectional study applied RRR to derive dietary patterns and showed that patterns involving high intake of phytonutrient-rich plant foods, tea, omega-3-rich deep-sea fish, and other protein-rich foods such as shellfish and milk exhibited an inverse dose-response association with frailty [16]. According to the study, the factor loading values obtained for the examined food items could be ordered as follows: fruit (−0.48), nuts and seeds (−0.39), tea (−0.34), vegetables (−0.33), whole grains (−0.27), shellfish (−0.23), milk (−0.21), and fish (−0.20) (16). We derived dietary patterns that comprised antioxidant drinks (tea), carbohydrates (rice), protein-rich foods (fish, meat, seafood, eggs, and milk), and phytonutrient-rich foods (fruit, dark green vegetables, other vegetables, and beans). We noted that more negative factor loading values for the food items were associated with lower frailty scores. The factor loading values for the food items could be ordered as follows: tea (−0.46), carbohydrates (−0.41), fruit (−0.40), fish (−0.35), meat (−0.33), seafood (−0.27), eggs (−0.23), and dark green vegetables (−0.21). The characteristics of the healthy dietary patterns observed in the present study are consistent with those reported by the aforementioned Taiwanese cross-sectional study [16], despite the differences in the definitions of frailty, the applied food frequency questionnaire, and factor loading values between these two studies. However, a cross-sectional study cannot be used to infer causation. In a cross-sectional design, determining whether participants’ dietary patterns contributed to their frailty or whether their frailty prompted them to adopt suitable dietary patterns is impossible [38].

The strengths of the present study are its longitudinal design, large sample, and inclusion of a broad range of sociodemographic characteristics, lifestyle factors, health status, mobility, and anthropometric measurements in the analyses. In this population-based cohort of elderly people, we observed the associations between dietary pattern scores and frailty status at baseline and could establish longitudinal associations between dietary patterns and frailty over 4-year, 8-year, and 12-year follow-up periods. In the future, we will attempt to explore a notable topic, namely the reversibility of healthy dietary patterns for individuals with pre-frailty or frailty. Despite the aforementioned strengths, this study has several limitations. The dietary assessment was conducted at a single time point, and whether participants maintained their dietary habits during the follow-up periods could not be ascertained. However, a previous study reported that a cohort of elderly people maintained their general dietary habits during follow-up [39]. Furthermore, the energy-based adjustment of dietary intake is usually important in epidemiologic analyses and dietary pattern research to evaluate the effects of nutrients. Nevertheless, relying upon existing data, we use body-size adjustment, including BMI, mid-arm circumference, calf circumference, and leg length instead of energy adjustment for our dietary pattern analysis of participants with geriatric syndrome. In addition, although we adjusted for potential confounding factors, the existence of unmeasured con-founders is conceivable. Additionally, the competing risk of frailty-free mortality and the short-term incidence of frailty might have led to the underestimation of the frailty incidence because our data were interval censored.

In this study, the RRR-derived patterns were based on limited food items. Whether the patterns derived from limited food items can be called a dietary pattern is a topic worth discussing. No clear definition is available on how many food items are required to indicate a dietary pattern. In any case, the RRR-derived pattern reflected the dietary characteristics (e.g., quantity) of a proportion of the Taiwanese population, and this pattern was related to their frailty status.

## 5. Conclusions

Our findings indicate that recommendations to increase the intake of antioxidant drinks (tea), energy-rich foods (e.g., carbohydrates such as rice), protein-rich foods (fish, meat, seafood, eggs, and milk), and phytonutrient-rich foods (fruit, dark green vegetables, other vegetables, and beans) could be inversely associated with the prevalence and incidence of frailty.

## Figures and Tables

**Table 1 ijerph-18-09717-t001:** Dietary patterns derived through reduced rank regression of food item data from the TLSA 1999 ^a^.

Food Items	1st Dietary Pattern: Factor Loading ^b^	2nd Dietary Pattern: Factor Loading ^c^
Tea	**−0.46**	**−0.27**
Carbohydrate	**−0.41**	−0.13
Fruit	**−0.40**	**0.20**
Fish	**−0.35**	**−0.26**
Meat	**−0.33**	**0.30**
Seafood (Fish not included)	**−0.27**	**0.42**
Egg	**−0.23**	−0.03
Deep-Green Vegetables	**−0.21**	0.08
Vegetables	−0.18	**0.56**
Milk	−0.12	**0.44**
Beans	−0.11	0.14
Percent Variation Accounted for RRR Factors (explained %)		
All food items (model effects)	15.10%	7.87%
Response variable (frailty score 1999, 2003, 2007, 2011)	8.02%	0.14%

TLSA, Taiwan Longitudinal Study on Aging; RRR, Reduced Rank Regression; ^a^ TLSA 1999, *n* = 3945. ^b,c^ Patterns were derived through RRR with frailty scores in 1999, 2003, 2007, and 2011 as the response variables and 11 foods items as the predictor variables; factor loadings with absolute values of ≥0.2 are shown in bold.

**Table 2 ijerph-18-09717-t002:** Baseline (1999) characteristics stratified according to frailty status in participants.

	Frailty Status (TLSA 1999, *n* = 3486)			
Characteristics	Non-Frailty	Pre-Frail	Frailty	*p*-Value
	*n* = 1500	*n* = 1512	*n* = 474	
	*n* (%)/	*n* (%)/	*n* (%)/	
	Mean ± SD	Mean ± SD	Mean ± SD	
** Socio-demographic variables**				
Age (years)				<0.0001
53–64	608 (48.3)	563 (44.7)	88 (7.0)	
65–74	586 (45.9)	514 (40.3)	176 (13.8)	
75+	306 (32.2)	435 (45.7)	210 (22.1)	
Gender				<0.0001
Female	501 (31.7)	782 (49.5)	296 (18.8)	
Male	1099 (52.4)	730 (38.3)	178 (9.3)	
Education				<0.0001
Illiterate	274 (27.4)	502 (50.2)	225 (22.5)	
Primary	701 (42.9)	734 (45.0)	198 (12.1)	
High School	376 (59.3)	219 (34.5)	39 (6.2)	
College and above	149 (68.4)	57 (26.2)	12 (5.5)	
Ethnicity				<0.0001
Fuchien	929 (40.4)	1036 (45.0)	337 (14.6)	
Hakka	262 (43.2)	268 (44.2)	76 (12.5)	
Mainlander	290 (53.1)	199 (36.5)	57 (10.4)	
Other	19 (59.4)	9 (28.1)	4 (12.5)	
Income				<0.0001
Unsatisfied	821 (37.5)	995 (45.4)	375 (17.1)	
Satisfied	679 (52.4)	517 (40.0)	99 (7.6)	
**Behavioral variables**				
Current smoker				<0.0001
No	1055 (40.4)	1158 (44.4)	396 (15.2)	
Yes	445 (50.7)	354 (40.4)	78 (8.9)	
Current alcohol use				<0.0001
No	969 (37.6)	1188 (46.1)	420 (16.3)	
Yes	531 (58.4)	324 (35.6)	54 (5.9)	
Exercise				<0.0001
Yes	1086 (57.4)	681 (36.0)	125 (6.6)	
No	414 (26.0)	831 (52.1)	349 (21.9)	
**Body measurements**				
BMI (kg/m^2^)	23.8 ± 3.2	23.5 ± 3.4	23.2 ± 4.0	0.0041
Upper arm Circumference (cm)	28.7 ± 3.5	28.3 ± 3.9	27.8 ± 4.3	<0.0001
Leg Circumference (cm)	34.9 ± 3.5	34.0 ± 3.7	32.5 ± 4.1	<0.0001
Leg length (cm)	45.2 ± 4.3	44.3 ± 4.6	43.9 ± 4.6	<0.0001
**Health status**				
Mobility function				<0.0001
Good	1161 (65.7)	581 (32.9)	25 (1.4)	
Impaired	339 (19.7)	931 (54.2)	449 (26.1)	
Number of diseases	0.7 ± 0.9	0.8 ± 1.0	1.3 ± 1.1	<0.0001

**Table 3 ijerph-18-09717-t003:** Baseline (1999) dietary characteristics stratified according to frailty status.

	Frailty Status (*n* = 3486)			
Dietary Characteristics	Non-Frailty	Pre-Frail	Frailty	*p*-Value
	*n* = 1500 (43.0%)	*n* = 1512 (43.4%)	*n* = 474 (13.6%)	
Daily meals				
≥3 meals	1461 (97.4)	1456 (96.3)	439 (92.6)	<0.0001
≤2 meals (ref)	39 (2.6)	56 (3.7)	35 (7.4)	
Food enough				
Yes	1498 (99.9)	1507 (99.7)	464 (97.9)	<0.0001
No (ref)	2 (0.1)	5 (0.3)	10 (2.1)	
Appetite good				
Yes	1445 (96.3)	1339 (88.6)	261 (55.1)	<0.0001
No (ref)	55 (3.7)	173 (11.4)	213 (44.9)	
Eat alone				
Yes	169 (11.3)	216 (14.3)	103 (21.7)	<0.0001
No (ref)	1331 (88.75)	1296 (85.7)	371 (78.3)	
Intake amount change				
No change	1462 (97.5)	1416 (93.7)	365 (77.0)	<0.0001
Change (ref)	38 (2.5)	96 (6.4)	109 (23.0)	
Eat less due to indigestion				
Yes	104 (6.9)	190 (12.6)	119 (25.1)	<0.0001
No (ref)	1396 (93.1)	1322 (87.4)	355 (74.9)	
Eat less due to disease				
Yes	311 (20.7)	403 (26.7)	164 (34.6)	<0.0001
No (ref)	1189 (79.3)	1109 (73.4)	310 (65.4)	

ref = reference group.

**Table 4 ijerph-18-09717-t004:** Association between dietary patterns and frailty status: cross-sectional (1999), Short-term (2003), medium-term (2007), and long-term (2011) effects.

	Model 1		Model 2		Model 3		Model 4	
	Pre-Frail vs. Non	Frailty vs. Non	Pre-Frail vs. Non	Frailty vs. Non	Pre-Frail vs. Non	Frailty vs. Non	Pre-Frail vs. Non	Frailty vs. Non
Dietary Group	ORp [CI]	ORf [CI]	ORp [CI]	ORf [CI]	ORp [CI]	ORf [CI]	ORp [CI]	ORf [CI]
Cross-sectional (*n* = 3486)								
General vs. Healthy	1.32 * [1.10,1.58]	1.78 ** [1.27,2.50]	1.31 * [1.10,1.57]	1.80 ** [1.29,2.52]	1.26 * [1.05,1.51]	1.55 * [1.09,2.21]	1.26 * [1.05,1.51]	1.55 * [1.09,2.21]
Unhealthy vs. Healthy	1.59 *** [1.31,1.94]	4.28 *** [3.10,5.90]	1.56 *** [1.28,1.89]	4.13 *** [2.99,5.71]	1.42 ** [1.16,1.74]	2.78 *** [1.97,3.93]	1.41 ** [1.15,1.72]	2.74 *** [1.94,3.87]
Short-term ^a^ (*n* = 2474)								
General vs. Healthy	1.26 [1.03,1.55]	1.89 * [1.26,2.83]	1.27 * [1.03,1.56]	1.91 * [1.28,2.87]	1.24 * [1.01,1.52]	1.83 * [1.22,2.76]	1.23 * [1.00,1.52]	1.78 * [1.18,2.69]
Unhealthy vs. Healthy	1.33 * [1.05,1.68]	2.86 *** [1.91,4.36]	1.35 * [1.07,1.71]	2.98 *** [1.98,4.51]	1.28 * [1.01,1.62]	2.74 *** [1.80,4.17]	1.26 [1.00,1.60]	2.55 *** [1.67,3.88]
Medium-term ^b^ (*n* = 1913)								
General vs. Healthy	1.09 [0.86,1.38]	1.44 ^†^ [0.98,2.12]	1.10 [0.88,1.39]	1.45 [0.99,2.14]	1.04 [0.83,1.32]	1.38 [0.93,2.03]	1.05 [0.83,1.33]	1.37 [0.93,2.03]
Unhealthy vs. Healthy	1.47 * [1.12,1.94]	1.77 * [1.15,2.73]	1.49 * [1.13,1.96]	1.79 * [1.16,2.76]	1.33 * [1.00,1.77]	1.68 * [1.08,2.61]	1.36 [1.03,1.80]	1.66 * [1.07,2.57]
Long-term ^c^ (*n* = 879)								
General vs. Healthy	1.34 [0.95,1.90]	1.30 [0.74,2.27]	1.32 [0.94,1.87]	1.39 [0.80,2.42]	1.31 [0.93,1.86]	1.34 [0.77,2.34]	1.26 [0.90,1.78]	1.17 [0.67,2.05]
Unhealthy vs. Healthy	1.84 * [1.21,2.78]	2.32 * [1.26,4.28]	1.89 * [1.25,2.86]	2.79 ** [1.52,5.15]	2.08 ** [1.36,3.19]	2.81 * [1.51,5.24]	1.81 * [1.20,2.74]	2.35 * [1.27,4.34]

Model 1 adjusted for demographic and lifestyle variables: age, gender, education, ethnicity, income/current economic status, smoking habits, and alcohol use; Model 2 adjusted for variables in Model 1 and added anthropometric variables from 1999 (body mass index, mid-arm circumference, calf circumference, and leg length); Model 3 adjusted for variables in Model 1 and added dietary characteristic variables from 1999 (daily meal frequency, food amount, appetite, whether the participant ate alone, changes in food intake amount, whether the participant ate less because of indigestion, constipation or diarrhea, whether the participant ate less because of diseases or medical orders); Model 4 adjusted for variables in Models 2 and 3 and added the variable concerning number of diseases; ^a^ Deducted those who had frailty in 1999; ^b^ Deducted those who had frailty in 1999 or 2003; ^c^ Deducted those who had frailty in 1999, 2003, or 2007; ^†^
*p* = 0.0673 (borderline significance); * *p* < 0.05, ** *p* < 0.001, *** *p* < 0.0001. Healthy = Healthy dietary pattern; General = General dietary pattern; Unhealthy = Unhealthy dietary pattern.

**Table 5 ijerph-18-09717-t005:** Sociodemographic, behavioral, dietary, anthropometric, health status, and food frequency characteristics of three dietary groups at baseline (year 1999).

		Dietary Groups			
Characteristics	Categories/Frequency/Unit	Healthy	General	Unhealthy	*p*-Value
		Mean ± SD /*n*(%)	Mean ± SD /*n*(%)	Mean ± SD /*n*(%)	
**Sociodemographic**					
Gender	Female	328 (27.1)	593 (50.7)	658 (59.5)	<0.0001
Age group	53–64	533 (44.0)	420 (35.9)	306 (27.6)	<0.0001
	65–74	419 (34.6)	417 (35.6)	440 (40.0)	
	75+	259 (21.4)	333 (28.5)	359 (32.4)	
Education level	Illiterate	202 (16.7)	344 (29.4)	455 (41.2)	<0.0001
	Primary	579 (47.8)	556 (47.5)	498 (45.1)	
	High school	324 (26.8)	190 (16.2)	120 (10.9)	
	College	106 (8.8)	80 (6.8)	32 (2.9)	
Current economic status	Satisfied	555 (45.8)	448 (38.3)	292 (26.4)	<0.0001
**Behavioral**					
Current smoker	Yes	410 (33.9)	245 (20.9)	222 (20.1)	<0.0001
Current alcohol use	Yes	443 (36.6)	252 (21.5)	214 (19.4)	<0.0001
Exercise	Yes	725 (59.9)	668 (57.1)	499 (45.2)	<0.0001
**Dietary Characteristics**					
Appetite	Poor	67 (5.5)	121 (10.3)	253 (22.9)	<0.0001
Intake amount change	Yes	47 (3.9)	59 (5.0)	137 (12.4)	<0.0001
Eat less (indigestion)	Yes	96 (7.9)	134 (11.5)	183 (16.6)	<0.0001
**Body Measurements**					
BMI	kg/m^2^	23.8 ± 3.2	23.7 ± 3.4	23.2 ± 3.6	0.0002
Leg circumference	cm	35.0 ± 3.6	34.1 ± 3.6	33.4 ± 3.9	<0.0001
**Health Status**					
Mobility	Impairment	416 (34.3)	617 (52.7)	686 (62.1)	<0.0001
Numbers of disease		0.7 ± 0.9	0.8 ± 0.9	1.0 ± 1.1	<0.0001
**Food Items**					
Carbohydrates	(bowls/day)	3.7 ± 1.4	2.9 ± 0.9	2.5 ± 0.8	<0.0001
Meat	<1 times/week	79 (6.5)	225 (19.2)	376 (34.0)	<0.0001
	1–2 times/week	160 (13.2)	247 (21.1)	316 (28.6)	
	≥3 times/week	972 (80.3)	698 (59.7)	413 (37.4)	
Fish	<1 times/week	51 (4.2)	113 (9.7)	330 (29.9)	<0.0001
	1–2 times/week	94 (7.8)	163 (13.9)	262 (23.7)	
	≥3 times/week	1066 (88.0)	894 (76.4)	513 (46.4)	
Seafood	< 1 times/week	659 (54.4)	837 (71.5)	942 (85.2)	<0.0001
	1–2 times/week	256 (21.1)	226 (19.3)	130 (11.8)	
	≥3 times/week	296 (24.5)	107 (9.2)	33 (3.0)	
Egg	< 1 times/week	186 (15.4)	261 (22.3)	416 (37.7)	<0.0001
	1–2 times/week	316 (26.1)	396 (33.9)	388 (35.1)	
	≥3 times/week	709 (58.6)	513 (43.9)	301 (27.2)	
Milk	< 1 times/week	442 (36.5)	464 (39.7)	562 (50.9)	<0.0001
	1–2 times/week	103 (8.5)	99 (8.5)	93 (8.4)	
	≥3 times/week	666 (55.0)	607 (51.9)	450 (40.7)	
Bean	< 1 times/week	273 (22.5)	329 (28.1)	384 (34.8)	<0.0001
	1–2 times/week	315 (26.0)	311 (26.6)	338 (30.6)	
	≥3 times/week	623 (51.5)	530 (45.3)	383 (34.7)	
Vegetable	<1 times/week	4 (0.3)	3 (0.3)	32 (2.9)	<0.0001
	1–2 times/week	5 (0.4)	11 (0.9)	42 (3.8)	
	≥3 times/week	1202 (99.3)	1156 (98.8)	1031 (93.3)	
Fruit	< 1 times/week	26 (2.2)	68 (5.8)	289 (26.2)	<0.0001
	1–2 times/week	48 (4.0)	83 (7.1)	267 (24.2)	
	≥3 times/week	1137 (93.9)	1019 (87.1)	549 (49.7)	
Tea	<1 times/week	314 (25.9)	814 (70.0)	944 (85.4)	<0.0001
	1–2 times/week	67 (5.5)	82 (7.0)	68 (6.2)	
	≥3 times/week	830 (68.5)	274 (23.4)	93 (8.4)	
Deep-green vegetable	≤2 times/week	72 (6.0)	137 (11.7)	257 (23.3)	<0.0001
	≥3 times/week	1139 (94.0)	1033 (88.3)	848 (76.7)	

## Data Availability

The datasets generated during the current study are not publicly available, but data are however available from the applicants upon reasonable request and with permission of the Ministry of Health and Welfare in Taiwan.

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
