# Peer review of "Cross-Sectional, Short-, Medium-, and Long-Term Effects of Dietary Pattern on Frailty in Taiwan"

_ijerph, 2021, doi:10.3390/ijerph18189717_

Round 1
Reviewer 1 Report
Dear authors,
First of all, I would like to thank you for preparing this research that evaluate the effects of dietary patterns on frailty.
Next, I will indicate some aspects that I think would improve the presented article.
- Paragraphs that start at line 80, line 89 and line 139 are the same paragraph.
- Statistical Methods is section 2.7 instead of 2.1. In addition, the content of this section should be revised because it gives the impression that a large part of the information contained would correspond to results.
- In Table 3, it would be advisable to include the P values as shown in Table 2.
- Check that reference numbers are correct because line 291 refers to a Spanish investigation and the reference number used does not match the one in the References section.
- The format of the bibliographic references must be adapted to the journal's specifications.
Kind regards,
Reviewer 2 Report
This study examined the association between dietary patterns and the frailty by conducting present cross-sectional study to identify frailty-related dietary patterns and their short-term, medium-term, and long-term effects on frailty development over a 12-year follow-up period in older adults in Taiwan. Several papers have done research on association between dietary patterns and frailty, while this study focus on traditional Asian diets which differ considerably from Western diets.
Comments:
Page 3 line 123: slow walking speed/slowness, having difficulty or being unable to walk a distance of 200 to 300 m is set as the criteria. However, this criteria is more like ability to walk instead of speed/slowness. If the criteria is set up as walking speed, a speed test should be performed or a time limit should be set up for walking a fixed distance.
Page 8 line 267: Participants in the T1 group “did not have smoking or drinking habits at the time of the study”. This is not consistent with the data in Table 5. The data shows percentage of current smoker and current alcohol use in T1 group is 33.9% and 36.69%, respectively. They are even higher than their percentage in T2 and T3 group (20.9% and 21.5%, 20.1% and 19.4%). Please explain.
Minor:
Page 5 Table 2: The number for Male Non-frailty is 99, which is not correct. Please double check.
Page 7 line 239: which model is discussed here? If it’s Model 4, the OR should be 1.23 for prefrailty instead of 1.24
Page 2 line 44-45 “proteins or amino acids constituting proteins” duplicate items, proteins are comprised of amino acids
Page 2 line 89-92 and Page 4 line 139-141: These two paragraphs are exactly the same as page 2 line 80-82. Duplicates need to be removed.
Page 10 line 291: incorrect reference. Should be reference #24
Page 10 line 296: incorrect reference. Should be reference #25
Reviewer 3 Report
The issue is important and topical to the journal. The study is well sized and the longitudinal design with many years of follow up is a plus. The introduction is clear. However, I am very much concerned about the statistical analyses and the derived conclusions, as detailed below.
- The authors apply RRR (Ref. 22). This method was introduced to identify dietary predictors (as assessed by dietary intake data) of a set of nutritional response variables (like intake of PUFAs or magnesium) which are thought to be biologically related to the outcome of interest (e.g. diabetes). In a certain way, you estimate how much “disease-critical” nutrients are consumed (based on prior knowledge/hypotheses on critical nutrients), and only in a second step you relate this pattern to an outcome. A study by Shin et al. 2018 from Korea exemplifies this point: food frequencies were related to estimated intake (=responses) of vitamin B6, vitamin C, and iron intakes, as these were considered to be important for MCI. The identified seafood and vegetables pattern (found to be strongly related to these response variables) was then related with MCI https://www.ncbi.nlm.nih.gov/pmc/articles/PMC5800199/
As far as I can understand, Shiau et al. did not apply the method this way, because they did not specify any nutritional response variables. Rather, they directly used the outcome of interest (frailty, assessed concurrently with diet and in the 2002, 2007, 2011 follow-ups) as the response variable). I think this is not a valid application of RRR. I may be wrong, but the authors would need to show, e.g. by citing other papers employing their approach, that RRR can be used that way. At least the rationale of using RRR, explained in ref. 22, does not apply to such a shortcut. Importantly, the cited references 23-25 either used an priori Food Pattern, or PCA-derived factors based on several hundred (!) food items assessed with FFQs.
If the four frailty scores at the different timepoints were used as response variables: how about loss-to follow-up and incomplete data sets?
- Even if so, the method seems to maximize the association of single food items with the outcome. The fist factor contains all food items, and is uniformly negatively associated with frailty. Does this imply more than showing that frail people eat and drink less? I understand that someone with a healthy food pattern drinks more tea, eats more rice, fish etc. As only these items were asked for, I do not understand how this can be an empirically derived pattern. To show the relevance of a pattern, overall caloric intake needs to be controlled, otherwise it suffices to say that frail subjects eat less of the typical Taiwanese food (which is assessed by the short list of 11 food items).
- The authors use causal language (“effects of dietary pattern on frailty”). How can causality be inferred, given that only associations are studied? The simultaneous use of all four time-points in the RRR, and the analysis of subjects at follow up who already were frail at baseline, does not allow to gauge the impact of a sufficient nutrition pattern on incident frailty.
In my view, the results would be easier to interpret if the association of each single food item with baseline frailty would be used to identify the most critical items (this will result in a similar list as factor 1), and – e.g. by using the weights from a logistic regression with baseline frailty as outcome – derive a weighted frailty related pattern. In a second step, the risk of becoming frail during follow could be associated with this weighed diet score, analyzing only those who were not frail at baseline.
The discussion would need to deemphasize the issue of patterns, as the authors simply lack the data do derive them empirically; rather they should focus on quantity, on the most important items related to frailty, and on the limitations of their method, which also precludes any comparison with the study from Hong Kong (Ref. 23).
Minor issues
Line 41: In summarizing previous literature, make clear that you cite recent systematic reviews and meta-anayses, not just some findings
Line 63: omit „cross-sectional“, because you report both cross-sectional and longitudinal associations. Or reformulate to reflect both aspects.
Line 130: typo, correct is: confounders
Round 2
Reviewer 3 Report
I still do not agree with the interpretation of dietary patterns (the method cannot identify them as distinct from food quantity). However, this may be more a matter of taste, and readers should be able to judge for themselves, whether a limited set of typical food items can ever be used for patterning. Therefore, I suggest only the following minor changes to get this paper over the line:
- make clear in the abstract that it is the higher intake of certain food items which is related to frailty, not the "pattern" per se: e.g. in line 11712, write "characterized by higher intake of green tea, energy-rich foods..."
- also in the abstract, make clear that the analyses are related to prevalence (at baseline) AND incidence (much more interesting) of new cases, e.g. by writing in line 24/25: "the unhealthy dietary pattern was associated with prevalent frailty (OR 2.74, CI = ??), and with short-term, medium-term, and long-term incident frailty (give ORs and CIs only for fraily, the pre-frailty associations are less relevant here and a distraction for the reader.)
- In general, when reporting ORs, you MUST report confidence intervals in the manuscript and abstract.
- I do not understand what exactly is shown in table 4. In cross-sectional associations, there is no T1 or T2 or T3. Short Term associations apparently associate incident cases at T1 (first follow up) with diet at baseline, but what do T3 vs. T1 mean here? Same problem for medium and long term. Please check.
- Also, the regression analysis needs to be better described in the statistical section, not only in a footnote. The statistical section only describes the food pattern analysis in detail, but the logistic regression section needs to be expanded. Which samples were analysed in the cross-sectional and longitudinal analyses, who was removed in the longitudinal analyses, which covariates were entered in the logistic regression?
